# Impacts of Arbuscular Mycorrhizal Fungi on Metabolites of an Invasive Weed *Wedelia trilobata*

**DOI:** 10.3390/microorganisms12040701

**Published:** 2024-03-29

**Authors:** Xinqi Jiang, Daiyi Chen, Yu Zhang, Misbah Naz, Zhicong Dai, Shanshan Qi, Daolin Du

**Affiliations:** 1School of Agricultural Engineering, Jiangsu University, Zhenjiang 212013, China; j18280577017@163.com (X.J.); cdy1145991633@163.com (D.C.); zhangyu001125@163.com (Y.Z.); 2Institute of Environment and Ecology, School of the Environment and Safety Engineering, Jiangsu University, Zhenjiang 212013, China; misbahnaz.ray@yahoo.com (M.N.); daizhicong@163.com (Z.D.); 3Jiangsu Collaborative Innovation Center of Technology and Material of Water Treatment, Suzhou University of Science and Technology, Suzhou 215009, China; 4Jingjiang College, Jiangsu University, Zhenjiang 212013, China

**Keywords:** arbuscular mycorrhizal fungi (AMF), metabolomics, plant invasion, plant–microbe interactions

## Abstract

The invasive plant *Wedelia trilobata* benefits in various aspects, such as nutrient absorption and environmental adaptability, by establishing a close symbiotic relationship with arbuscular mycorrhizal fungi (AMF). However, our understanding of whether AMF can benefit *W. trilobata* by influencing its metabolic profile remains limited. In this study, Liquid chromatography-tandem mass spectrometry (LC-MS/MS) was conducted to analyze the metabolites of *W. trilobata* under AMF inoculation. Metabolomic analysis identified 119 differentially expressed metabolites (DEMs) between the groups inoculated with AMF and those not inoculated with AMF. Compared to plants with no AMF inoculation, plants inoculated with AMF showed upregulation in the relative expression of 69 metabolites and downregulation in the relative expression of 50 metabolites. AMF significantly increased levels of various primary and secondary metabolites in plants, including amino acids, organic acids, plant hormones, flavonoids, and others, with amino acids being the most abundant among the identified substances. The identified DEMs mapped 53 metabolic pathways, with 7 pathways strongly influenced by AMF, particularly the phenylalanine metabolism pathway. Moreover, we also observed a high colonization level of AMF in the roots of *W. trilobata*, significantly promoting the shoot growth of this plant. These changes in metabolites and metabolic pathways significantly affect multiple physiological and biochemical processes in plants, such as free radical scavenging, osmotic regulation, cell structure stability, and material synthesis. In summary, AMF reprogrammed the metabolic pathways of *W. trilobata*, leading to changes in both primary and secondary metabolomes, thereby benefiting the growth of *W. trilobata* and enhancing its ability to respond to various biotic and abiotic stressors. These findings elucidate the molecular regulatory role of AMF in the invasive plant *W. trilobata* and provide new insights into the study of its competitive and stress resistance mechanisms.

## 1. Introduction

The interaction between soil microorganisms and plant root systems is ubiquitous, with the symbiotic relationship between plants and arbuscular mycorrhizal fungi (AMF) representing the oldest and most widespread form of interaction between plants and fungi [1]. Within the plant, this interaction is primarily achieved through mycorrhiza-mediated nutrient transfer [2], enhancement of plant stress tolerance, and alterations in plant transcriptome and metabolome [3,4]. Outside the plant, AMF can indirectly influence the growth and development of plants by regulating microbial community structure and soil structure [2,5,6], utilizing an extensive mycelial network to modulate soil nutrients [5].

*Wedelia trilobata* (L.) Hitchc., also known as *Sphagneticola trilobata* (L.) Pruski, is a perennial clonal herb in the Asteraceae family [7] and is primarily native to Central and South America. This plant has been introduced extensively as an ornamental, medicinal, and ground cover species in numerous tropical and subtropical regions worldwide and is now widely distributed across Asia, Africa, and the Pacific, such as Southeast Asia, South Asia, East Africa, and other tropical and subtropical climate zones [8]. Due to its rapid growth and adaptability, *W. trilobata* exhibits strong invasive tendencies [9], posing threats to biodiversity, ecological stability, and agriculture in invaded regions. As such, many countries recognize it as an aggressive and destructive invader [10]. Consequently, *W. trilobata* has been recognized by many countries as an extremely aggressive and destructive plant and is listed by the International Union for Conservation of Nature (IUCN) among the world’s 100 worst invasive alien species [8].

The symbiotic association between plants and fungi is widely recognized as a key factor influencing and driving the success of plant invasions [11,12,13]. Studies have indicated that invasive Asteraceae plants exhibit stronger mutualistic interactions with AMF than their phylogenetically related native counterparts [14]. This interaction significantly enhances the competitiveness of invasive Asteraceae plants against native plants, resulting in increased total biomass, heightened stress tolerance, and absorption of nutrients, mainly phosphorus (P). For example, AMF can assist the invasive plant *Solidago canadensis* in acquiring P in P-deficient environments, thereby altering its resource allocation strategy to gain a growth advantage in the shoot [15]. In nutrient-deprived environments, AMF has been observed to facilitate the growth of *W. trilobata* while concurrently enhancing its resistance to pathogens [16]. These AMF-induced changes reflect the influence of AMF on invasive plants, potentially offering a partial explanation for the successful invasion of exotic plants. However, the precise mechanisms through which AMF affects invasive plants remain largely unknown, and the specificity of plant-fungus symbiosis outcomes further complicates the explanation of this mechanism [17,18,19].

Metabolomics is a powerful tool for capturing the physiological status, stress responses, and metabolic changes in plants under the influence of both biological and abiotic factors [20]. It is widely applied in the fields of plant and microbial research [21,22,23]. Metabolites are the ultimate products of physiological regulation in plants. Changes in their types and levels reflect a plant’s responses to genetic and environmental factors. Therefore, metabolic profiling analysis is essential to decipher plant growth, metabolic pathways, and stress response mechanisms. In recent years, many scholars have applied metabolomics technology, especially untargeted metabolomics, to study plant-fungus interaction systems [22,24,25,26]. The diverse effects of AMF on plants are due to metabolites produced by both symbiotic partners. AMF colonization reprograms plant metabolic pathways, altering primary and secondary metabolite levels. Through these metabolic changes, AMF regulates their host plants’ growth, development, and ability to respond to stress [27,28].

In recent years, researchers have focused on understanding the biological characteristics, invasive mechanisms, and ecological impacts of *W. trilobata* through in-depth investigations of its genetic traits, physiological features, and environmental adaptability [16,29,30]. However, the metabolic characteristics of this invasive plant have received little attention. To our knowledge, current research on the metabolomics of invasive *Wedelia trilobata*, as well as the impact of AMF on its invasion process from a metabolomics perspective, is limited. This study aims to identify and quantify metabolic differences in *W. trilobata* under mycorrhizal and non-mycorrhizal conditions. The obtained metabolomic data are expected to elucidate whether the impact of AMF on invasive *W. trilobata* can be explained from a metabolomics standpoint. This study also seeks to reveal the molecular mechanisms through which AMF facilitates the establishment and spread of invasive plant species. Additionally, it provides essential foundational data and a theoretical basis for formulating effective control strategies against this invasive species.

## 2. Materials and Methods

### 2.1. Study Species

Stem segments of *W. trilobata* were collected from Xia’men City, Fujian Province, China (119°31.76′ E, 32°12.02′ N), and propagated in a greenhouse at Jiangsu University. To ensure experimental accuracy, stem segments with similar growth status and good health were selected and subjected to surface sterilization with a 5% sodium hypochlorite solution for 10 min, followed by five rinses with sterile water. Each plant was maintained with two stem nodes during the experimental planting process. AMF species, *Glomus versiforme* (GV), was obtained from the Bank of Glomeromycota at the Institute of Plant Nutrition and Resources, Beijing Academy of Agriculture and Forestry Sciences (Beijing, China). The GV inoculum was a self-propagated mixed inoculum containing hyphae, AMF spores, and infected sorghum root tissue, with approximately 23 spores per gram of GV inoculum.

### 2.2. Experimental Design

The experiment was conducted using a sand culture potting method with sterilized river sand (diameter < 2 mm, devoid of any available nutrients) as the growth medium. The sand was placed in circular plastic pots (9 cm × 6 cm × 7.5 cm), and the experimental treatments included two AMF treatments: (1) no AMF inoculation, with 300 g of sand as the growth medium (control treatment, CK); and (2) AMF inoculation, with 6 g of GV fungal inoculum, mixed uniformly with 294 g of sand as the growth medium (AMF treatment, GV). Each pot was planted with one *W. trilobata* cutting, and a total of 10 pots (5 replicates for each AMF treatment) were used. The plant cuttings were inserted vertically in the center of each pot, with one node buried in the sand. All plants were cultured in a greenhouse with 70% relative humidity, a temperature of 25 °C, and a light period of 16/8 h day/night. The plants were watered with distilled water at appropriate intervals of 2 days. To supplement nutrients required for plant growth, 50 mL of low-P 1 × Hoagland solution was added to each plant every 5 days, containing 1.545 mg/L P with the form of PO_4_^3−^. After two months of growth, all the plants were harvested, and the growth parameters were measured, including mycorrhization frequency, stem length, root length, shoot dry biomass, and root dry biomass. Leaf samples were sent to the company (Suzhou PANOMIX Biomedical Tech Co., Ltd., Suzhou, China) for metabolomic analysis.

### 2.3. Determination of Mycorrhization Frequency

Plant root staining was conducted following the methods of Phillips and Hayman [31]. The staining process involved the following steps: suitable young roots were selected and washed before being cut into 2 cm segments, which were then digested with 10% KOH, rinsed with 30% H_2_O_2_, acidified with 1% HCl, stained with 0.05% Trypan Blue, and decolorized with 50% lactic acid. The fungal structures, such as hyphae, arbuscules, and vesicles, were immediately observed under the microscope to assess the AMF colonization levels of each sample and ultimately calculate the average mycorrhization frequency. The mycorrhization frequency was calculated using the formula:$$Mycorrhization\;frequency\left(\%\right)=\frac{Number\;of\;colonized\;root\;segments}{Total\;number\;of\;root\;segments\;observed}\times 100\%$$

### 2.4. Metabolomic Analysis

#### 2.4.1. Metabolite Extraction

We concurrently collected the second pair of fully expanded leaves from the apex of each plant in both treatment groups for metabolomic analysis. The metabolomic results were used to compare the metabolic differences between AMF-inoculated and non-AMF-inoculated *W. trilobata*.

In the experimental procedure, 200 mg (±1%) of the leaf sample was accurately weighed and placed in a 2 mL EP tube, followed by the addition of 0.6 mL of 2-chlorophenylalanine (4 ppm) in methanol (−20 °C), with vortexing for 30 s. Subsequently, 100 mg of glass beads were added, and the samples were subjected to grinding in the TissueLysis II tissue grinding machine at 25 Hz for 60 s. Following this, ultrasound treatment was carried out at room temperature for 15 min. The processed samples were then centrifuged at 25 °C for 10 min at 1750× *g*, and the supernatant was filtered through a 0.22 µm membrane to obtain the prepared samples for LC-MS. To establish quality control, 20 µL aliquots were taken from each sample for the preparation of quality control (QC) samples. The remaining samples were utilized for LC-MS detection.

#### 2.4.2. Chromatographic Conditions

In the Thermo Ultimate 3000 system, chromatographic separation was conducted using a column (ACQUITY UPLC^®^ HSS T3, 150 × 2.1 mm, 1.8 µm, Waters) maintained at 40 °C. The autosampler temperature was set to 8 °C. Analyte elution employed a gradient with either 0.1% formic acid in water (C) and 0.1% formic acid in acetonitrile (D) or 5 mM ammonium formate in water (A) and acetonitrile (B), at a flow rate of 0.25 mL/min. Post-equilibration, 2 μL injections of each sample were administered. A linear gradient of solvent B (*v*/*v*) was employed in the following manner: 0~1 min, 2% B/D; 1~9 min, 2%~50% B/D; 9~12 min, 50%~98% B/D; 12~13.5 min, 98% B/D; 13.5~14 min, 98%~2% B/D; 14~20 min, 2% D-positive model (14~17 min, 2% B-negative model).

#### 2.4.3. Mass Spectrometric Conditions

Thermo Q Exactive was used for the execution of ESI-MSn experiments, enabling operation in both positive and negative ionization modes. The positive ionization spray voltage was adjusted to 3.80 kV, while the negative ionization spray voltage was set to 2.50 kV. The sheath gas maintained a flow rate of 30 arbitrary units (arb), and the auxiliary gas was set at 10 arb. The capillary temperature was fixed at 325 °C, employing a resolution of 70,000 for a comprehensive scan across the mass range of m/z 81–1000. With the HCD scan, data-dependent acquisition (DDA) MS/MS experiments were conducted. The normalized collision energy was 30 eV. Dynamic exclusion was implemented to eliminate redundant MS/MS data.

#### 2.4.4. Data Processing and Multivariate Data Analysis

The raw data transformation was conducted through Proteowizard software (v3.0.8789), coupled with the application of the XCMS package in R (v3.3.2) for comprehensive sample data preprocessing. This encompassed tasks such as peak identification, peak filtration, and peak alignment. Subsequent multivariate statistical analyses of the metabolomic data were performed using the R language, employing the ropls package. These analyses included Principal Component Analysis (PCA), Partial Least Squares-Discriminant Analysis (PLS-DA), and Orthogonal Partial Least Squares Discriminant Analysis (OPLS-DA). Differential metabolites were filtered based on preset criteria in statistical tests, including a *p* value < 0.05 and a Variable Importance in Projection (VIP) of the first principal component in OPLS-DA > 1. The experiment employed MetPA, a KEGG-based metabolomics pathway analysis tool, to conduct pathway enrichment analysis on differentially expressed metabolites (DEMs). The MetPA database utilizes metabolic pathway enrichment and topological analysis to identify metabolic pathways that may be perturbed by AMF treatment, enabling further analysis and the creation of a metabolic pathway impact factor plot.

## 3. Results

### 3.1. Root Colonization and Plant Growth

Microscopic staining analysis revealed an average mycorrhization frequency of 81.03% (±6.45%) across the examined plants. Well-developed arbuscules, vesicles, and mycelial networks were observed in the roots of each sample treated with AMF. No obvious fungal structures were observed in the roots of plants that were not treated with AMF.

The inoculation with GV strain significantly increased stem length and shoot dry biomass of *W. trilobata*, but had no significant effect on root length and root dry biomass accumulation (Figure 1).

### 3.2. Metabolomics Analysis

#### 3.2.1. Multivariate Statistical Analysis

During metabolomics research based on mass spectrometry, it is customary to subject metabolomic datasets, pre-screened through rigorous QC and quality assurance (QA) protocols, to the analytical techniques of PCA and OPLS-DA. PCA showed that all samples were located within the 95% confidence interval and were effectively classified into two groups: the control group (CK) and the AMF treatment group (GV) (Figure 2a,b). Further analysis of the mass spectrometry data using OPLS-DA revealed significant differentiation between the control and AMF treatment groups on the first principal component axis, with the two groups located on the left and right of the confidence interval, respectively (Figure 2c,d). The results of multivariate statistical analysis showed that our assessment model possessed good quality, effectiveness, and reliability, with a relatively strong predictive capability.

#### 3.2.2. Comprehensive Identification of Metabolites

In the process of metabolite identification, the initial step involves the verification of precise molecular weights of metabolites (with a molecular weight error < 15 ppm). Subsequently, fragmentary information derived from MS/MS patterns is utilized for further annotation and matching in databases such as Metlin (http://metlin.scripps.edu; accessed on 23 March 2020) and MoNA (https://mona.fiehnlab.ucdavis.edu; accessed on 23 March 2020). A total of 368 metabolites were successfully annotated and categorized into nine distinct classes based on their functionalities. Among these metabolites, there were 116 metabolites in amino acid category, 80 metabolites in lipid category, 54 metabolites in carbohydrate category, 34 metabolites in cofactors and vitamins category, 23 metabolites in nucleotide category, 18 metabolites in xenobiotics category, 3 metabolites in energy category, only 1 metabolite in peptide category, and 39 metabolites that have not been identified in the unknown category (Figure 3).

#### 3.2.3. Differential Metabolite Analysis

Further selection was conducted based on the *P* value, Variable Importance in Projection (VIP) of the first principal component in OPLS-DA. Metabolite molecules were considered statistically significant when the *p*-value was <0.05 and VIP > 1. Ultimately, 119 differential metabolites were identified between the AMF treatment group and the non-AMF treatment group. In comparison to the non-AMF treatment group, 69 metabolites showed upregulated relative expression in the AMF treatment group, while 50 metabolites exhibited downregulated relative expression (Figure 4).

Hierarchical clustering analysis was conducted based on the metabolic levels of metabolites from *W. trilobata* under different AMF treatments. The dataset was scaled using the pheatmap package in R (v3.3.2) to generate a hierarchical clustering heatmap of relative quantification values of metabolites. In this study, we selected the categories with the top five quantities of differentiating metabolites, encompassing amino acids, carbohydrates, lipids, cofactors and vitamins, and nucleotides. Additionally, to visually illustrate the differential metabolite expression levels between the two AMF treatment groups within different substance categories, we have chosen several key differential metabolites from each category for presentation. Among the amino acid category (Figure 5a), 41 differential metabolites were detected, including 22 upregulated and 19 downregulated metabolites. AMF colonization significantly enhances the expression levels of L-proline, γ-aminobutyric acid, L-phenylalanine, and L-serine in *W. trilobata* (Figure 5b–e).

In the carbohydrate category, a total of 18 differential metabolites were detected (Figure 6a). Within these differential metabolites, fumaric acid and 6-phosphogluconic acid exhibits about twice as much as that in AMF inoculation treatment compared to non-inoculated treatment (Figure 6b,c).

Nineteen differential metabolites were detected in the lipid category (Figure 6d). GV inoculation notably resulted in increased levels of (S)-abscisic acid (Figure 6e) and certain flavonoid compounds such as luteolin (Figure 6f) in *W. trilobata*.

A total of 16 differential metabolites were detected in the category of cofactors and vitamins (Figure 7a), including 11 upregulated substances involved in pathways such as riboflavin metabolism (such as flavin mononucleotide (FMN) and riboflavin) (Figure 7b,c), nicotinate and nicotinamide metabolism, porphyrin and chlorophyll metabolism, and vitamin B6 metabolism. Five downregulated metabolites were mainly related to the biosynthesis of ubiquinone and other terpenoid quinones, as well as folic acid biosynthesis.

Six differential metabolites were detected in nucleotide category (Figure 7d), with a significant upregulation observed in the levels of allantoic acid and a marked decrease in cyclic adenosine monophosphate (AMP) levels (Figure 7e,f).

### 3.3. Metabolic Pathway Analysis

Metabolic pathways depict a series of ordered biochemical reactions within a plant that involve specific substrates. These pathways encompass many crucial life processes, including photosynthesis, respiration, sugar metabolism, nitrogen metabolism, and others. Analysis of metabolic pathways for differential metabolites reveals that 119 differential metabolites were mapped to 53 metabolic pathways. The bubble points located closer to the upper-right corner of the metabolic pathway impact factor bubble chart represent the metabolic pathways significantly influenced by AMF (Figure 8). Therefore, metabolic pathways more prone to the influence of AMF influence include phenylalanine metabolism, alanine, aspartate, and glutamate metabolism, arginine and proline metabolism, glutathione metabolism, β-alanine metabolism, isoquinoline alkaloid biosynthesis, as well as pantothenate and CoA biosynthesis. The seven metabolic pathways and the differential metabolites detected within these pathways are detailed in Table 1. Among these pathways, phenylalanine metabolism is the most significantly impacted, with a −log(P) of 3.4903 and a pathway impact score of 0.66667. Among the seven metabolic pathways, the pathway with the highest abundance of differentially accumulated metabolites is associated with arginine and proline metabolism, which contains seven differential metabolites.

## 4. Discussion

### 4.1. Effects of AMF on Invasive Plant Growth

In this study, in comparison with previous studies of other plant species under AMF treatment, our experiment observed a high level of AMF colonization in the roots of *W. trilobata* [15,32]. Symbiosis with AMF also resulted in a significant increase in the biomass and stem length of *W. trilobata*. Numerous studies have consistently reported the capacity of AMF to enhance plant growth, and these studies confirm that this phenomenon is closely linked to the more efficient utilization and absorption of water and nutrients facilitated by AMF [33,34]. The formation of mycorrhizal networks enables plants to acquire a broader range of nutrients through AMF. This sharing mechanism is particularly pronounced in nutrient-poor or stressful soil environments [15,35]. Consequently, plants colonized by AMF typically exhibit higher biomass in both shoots and roots. In the context of invasive plants, the growth-promoting influence of AMF remains significant. For instance, AMF can stimulate the P absorption of the invasive clonal plant *Solidago canadensis*, thereby promoting its shoot growth [15]. Similarly, studies on the invasive plant *Microstegium vimineum* indicate that symbiosis with AMF significantly increases biomass, creeping stem, and the number of aerial roots, thereby enhancing the competitive advantage of *M. vimineum* expansion [36,37]. The growth-promoting effects of AMF were also evident in this study, as demonstrated by an increase in both biomass and stem length of *W. trilobata*. Increased biomass potentially enables *W. trilobata* to occupy more resources in the ecosystem, including light, water, and nutrients, implying faster growth and reproduction rates as well as a stronger competitive advantage [38]. The increase in stem length may also suggest the presence of more growth points, aiding in the efficient occupation of ecological niches, providing opportunities for widespread dispersal, and accelerating population expansion and the establishment of new invasive areas [39].

### 4.2. Regulation of Plant Metabolites by AMF

Extensive research indicates that the initial colonization of plants by AMF activates the plant’s immune response and induces a state of defense readiness without incurring high adaptive costs that would impact growth [40]. Upon perceiving AMF, plants may undergo alterations at the physiological, transcriptional, and metabolic levels, constituting the initiation phase. Subsequently, when confronted with environmental stressors, plants will be more effective at mounting faster, stronger, and more enduring defense responses to the challenges [41]. Therefore, the low-cost immune priming induced by AMF not only avoids hindering proactive plant growth but also enhances the plant’s ability to respond to biotic and abiotic stressors to some extent. Our research findings provide the first direct evidence that AMF alters the metabolites of the invasive weed *W. trilobata*. Consequently, these changes in metabolites may play a vital role in the growth and stress resistance of this invasive weed.

#### 4.2.1. Impacts of Differential Metabolites on Plant Growth

In this study, the level of allantoic acid in *W. trilobata* inoculated with AMF was significantly increased. Allantoic acid is a high-nitrogen compound, and its accumulation promotes the efficient utilization of nitrogen in plants to some extent. Allantoin and its primary derivative, allantoic acid, play crucial roles in nitrogen assimilation and metabolism. In leguminous plants, allantoin and allantoic acid are regarded as primary forms of nitrogen transport. In non-leguminous plants, allantoin is still considered a potential nitrogen source under nitrogen-limiting conditions [42,43]. Studies indicate that under nitrogen-limiting conditions, allantoin can serve as a supplementary nitrogen source for rice, and the accumulation of allantoin supports plant growth [44]. Therefore, the accumulation of nitrogen-containing substances, such as amino acids and organic acids, observed in this study may also be related to the increased level of allantoic acid. This suggests that AMF may redistribute nitrogen allocation in *W. trilobata*, improving its availability and promoting the synthesis of various nitrogen-containing compounds and overall biomass accumulation.

The exchange of substances between plants and AMF is critical in maintaining the symbiotic relationship. AMF provides nutrients to plants through mycorrhizal tissues, while plants produce carbon through photosynthesis to supply the fungi. In this process, plants bear 4–20% of the carbon cost [45]. The carbon demand of AMF implies an additional carbon output from plants, thereby influencing plant carbon metabolism and allocation [46]. Carbon metabolism is the fundamental biochemical process in which organisms fix, transform, and utilize carbon compounds. Carbon storage in this metabolism is critically important, playing a vital role in energy regulation and organic compound synthesis. Our research reveals that symbiosis with AMF significantly increases the fumaric acid content in *W. trilobata*. Some studies reported that fumarate may serve as a potential flexible carbon sink in plant photosynthesis. In some cases, the carbon accumulated as fumarate can even be comparable to the carbon stored as starch [47]. Therefore, in the symbiotic association between AMF and plants, plants may opt to store a portion of carbon in fumaric acid to meet their own growth and additional AMF carbon requirements. This suggests that AMF may influence plant carbon metabolism, including carbon storage forms. This flexible carbon reservoir may further enhance plant biomass and contribute to better adaptation to changing environments. Additionally, research indicates that organic acid content in rapidly growing plants is significantly higher than in slow-growing plants, suggesting that high concentrations of organic acids may be more favorable for plant growth [48]. Consequently, in our study, the significant increase in biomass of *W. trilobata* may be closely related to the substantial accumulation of fumaric acid.

#### 4.2.2. Impacts of Differential Metabolites on Plant Stress Resistance

In this study, the inoculation with AMF significantly enhances the level of amino acid metabolism in *W. trilobata*, leading to the accumulation of key amino acids, such as L-proline, L-phenylalanine, and γ-aminobutyric acid (GABA). Currently, a wealth of research has demonstrated the essential role of these amino acid substances in enhancing plant resistance to environmental stressors [49,50,51].

Proline, serving as a critical regulator, plays an important role in augmenting plant resistance to diverse abiotic stresses [52,53,54]. AMF can enhance proline metabolism in rice under low-temperature and low-nitrogen conditions to cope with environmental stress [55]. Al-Arjani et al. [49] also reported that AMF induces the accumulation of proline in *Ephedra foliata*, contributing to improved tolerance to drought stress. Similar research indicates that the accumulation of proline in mycorrhizal symbiotic systems is a responsive mechanism of plants mediated by AMF to external environmental stress. In our study, AMF-mediated accumulation of proline in *W. trilobata* was observed, which may be indicative of an enhanced ability to resist environmental stress in its invaded habitats.

L-phenylalanine participates in the phenylpropanoid pathway as a precursor of phenylpropanoid compounds, such as lignin, anthocyanins, phenolic acids, and flavonoids [27,56], which significantly affect various physiological and biochemical processes in plants, including defense, nutrient absorption, and signal transduction [57,58]. Under salt stress conditions, the phenylpropane pathway is upregulated under the influence of AMF [50]. Similarly, the resistance of mycorrhizal tomato plants to the tomato mosaic virus is associated with an elevated level of flavonoids under mycorrhizal conditions [59]. Furthermore, L-phenylalanine is also implicated in the establishment of plant-fungi symbiosis, primarily through the biosynthesis of flavonoids [60,61]. In comparison to non-mycorrhizal plants, mycorrhizal plants often exhibit higher levels of flavonoids [62]. Our experimental results demonstrate that inoculation with AMF significantly enhances L-phenylalanine metabolism in *W. trilobata* and elevates levels of certain flavonoids, such as luteolin and apigenin. These results suggest that AMF may activate the phenylpropanoid pathway by reorganizing intermediates like L-phenylalanine, resulting in accumulated phenylpropanoid compounds. This could enhance stress tolerance in *W. trilobata* and strengthen the plant-fungi symbiosis establishment.

γ-aminobutyric acid (GABA) is a stress-induced product in plants [63] and serves as a crucial element in the plant’s response to diverse environmental stressors [64,65,66,67]. For instance, Invasive plants can enhance their immune response against pathogens by regulating the balance of regulating reactive oxygen species via GABA [68]. Additionally, the significant accumulation of GABA in the leaves of *Calopogonium mucunoides* resulting from mycorrhization is also considered one of the factors mitigating the heavy metal Pb toxicity [51]. Furthermore, GABA can also be transported to mitochondria, where it is converted by GABA transaminase and succinic semialdehyde dehydrogenase, ultimately producing succinate, a substrate in the tricarboxylic acid cycle [69]. This process participates in cellular respiration and carbon metabolism, which is crucial for plant growth and development under carbon or nitrogen limitation conditions [70,71]. In this study, we observed the accumulation of GABA and three metabolic pathways significantly impacted by AMF, including arginine and proline metabolism, beta-alanine metabolism, and aspartate, alanine, and glutamate metabolism. Previous research has highlighted the importance of these three metabolic pathways in facilitating GABA accumulation [72]. Therefore, our findings collectively confirm the positive effects of AMF on the synthesis of GABA in *W. trilobata*.

The establishment, maturation, defense, and acceptance of plant-fungal symbiosis, as well as achieving the delicate balance between symbiosis and immunity to maximize mutual benefits, is a complex and dynamic process. The evolving symbiotic relationship also drives adjustments in plant and fungal defense and symbiotic strategies [73]. For example, plants produce a range of secondary metabolites to resist biotic disturbance (including fungi) or enhance symbiotic mutualism upon sensing the presence of fungi [74]. Research indicates that plants exhibit a phenomenon termed “symbiotic regulation” in response to AMF. Throughout the entire process, from initial colonization to the establishment of stable symbiosis, plants rigorously regulate their defense mechanisms to mitigate carbon loss resulting from excessive colonization by AMF [75]. For instance, during the early stages of AMF colonization, plants induce the rapid accumulation of defensive substances such as salicylic acid to inhibit AMF colonization. However, this inhibition is temporary and does not affect the final colonization status [76]. Subsequently, after the establishment of colonization, the accumulation of defensive hormones such as jasmonic acid and abscisic acid is observed [77]. The temporal variations in hormone levels underscore the diverse responses of plant hormones to AMF. In this study, we indeed observed a significant upregulation of abscisal aldehyde, abscisic acid glucose esters, (S)-abscisic acid. It is known that abscisal aldehyde can be oxidized to form abscisic acid, and abscisic acid glucose esters can serve as a form of storage or long-distance transport of abscisic acid [78,79]. These findings indicate that inoculation with AMF can significantly enhance abscisic acid metabolism levels in *W. trilobata*. The synthesis may play a dual role: on one hand, potentially governing the degree of the symbiotic association between *W. trilobata* and AMF; on the other hand, concurrently enhancing *W. trilobata*’s capacity to respond to external biological and abiotic stresses.

In summary, our experiment evidenced a high mycorrhization frequency formed by AMF in *W. trilobata*. We discovered that within this symbiotic system, AMF rearranges the metabolic pathways of *W. trilobata*, resulting in advantageous outcomes for the plant. The mutual benefits are ultimately manifested in two key aspects. First, mycorrhizae-mediated mechanisms enable the rearrangement of carbon metabolism and efficient nitrogen utilization, leading to increased plant dry biomass and stem elongation. Second, AMF may potentially induce defense mechanism initiation in *W. trilobata* by altering its metabolic profile. This results in accumulated amino acids, certain plant hormones, and flavonoid compounds, enhancing the plant’s stress resistance. These mycorrhizae-induced metabolic changes may improve invasive *W. trilobata*’s environmental stress tolerance and competitiveness. This could allow rapid and stable occupation of ecological niches, facilitating successful invasion.

## 5. Conclusions

This study revealed that AMF can promote the growth of the invasive weed *W. trilobata*. For the first time, untargeted metabolomics was employed to unveil the impact of AMF on the metabolites and metabolic pathways of *W. trilobata*. Under AMF mediation, *W. trilobata* demonstrated elevated levels of amino acids, organic acids, flavonoids, and plant hormones, especially substances such as L-proline, L-phenylalanine, γ-aminobutyric acid, luteolin, apigenin, allantoic acid, fumaric acid, and abscisic acid. The accumulation of these metabolites may enhance *W. trilobata*’s ability to respond to various biotic and abiotic stresses, thereby conferring growth and competitive advantages. These findings elucidate the significant influence of AMF on *W. trilobata* from a metabolomics perspective, emphasizing AMF’s crucial role in driving the successful invasion of *W. trilobata*.

## Figures and Tables

**Figure 1 microorganisms-12-00701-f001:**
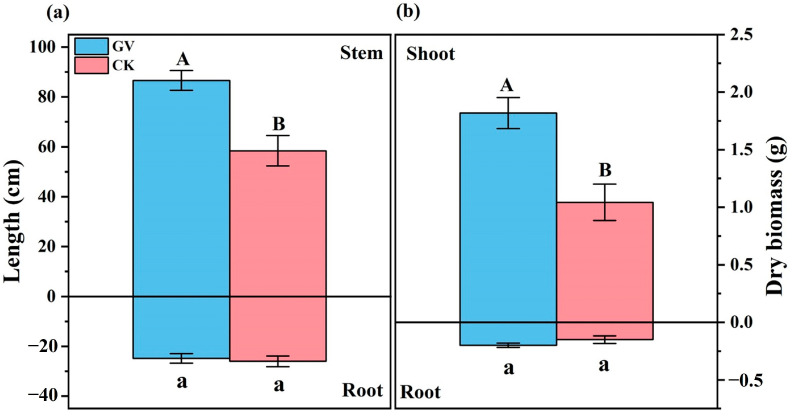
The stem length, root length (**a**), and shoot and root dry biomass (**b**) of *W. trilobata* under conditions of AMF inoculation (*Glomus versiforme*, GV) and no AMF inoculation (CK). Different letters indicate significant differences between treatments with uppercase letters for above-ground and lowercase letters for below-ground.

**Figure 2 microorganisms-12-00701-f002:**
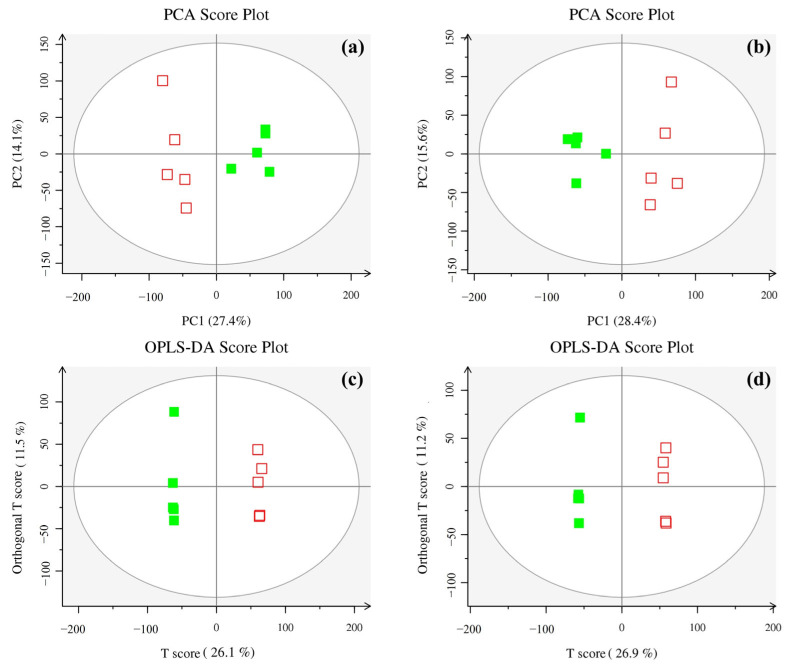
PCA scores plot for principal component analysis (PCA) in positive (**a**) and negative modes (**b**), and the scores plot for orthogonal projections to latent structures discriminant analysis (OPLS-DA) in positive (**c**) and negative modes (**d**). The *x*-axis denotes the explanatory power of the first principal component, and the *y*-axis denotes the explanatory power of the second principal component. Data points refer to experimental samples, green squares refer to the treatment of AMF inoculation (*Glomus versiforme*, GV), while red squares represent the treatment of no AMF inoculation (CK).

**Figure 3 microorganisms-12-00701-f003:**
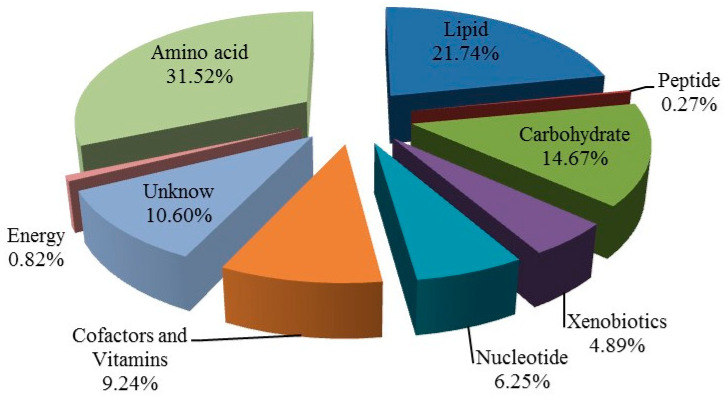
Classification chart of comprehensive identification of metabolites.

**Figure 4 microorganisms-12-00701-f004:**
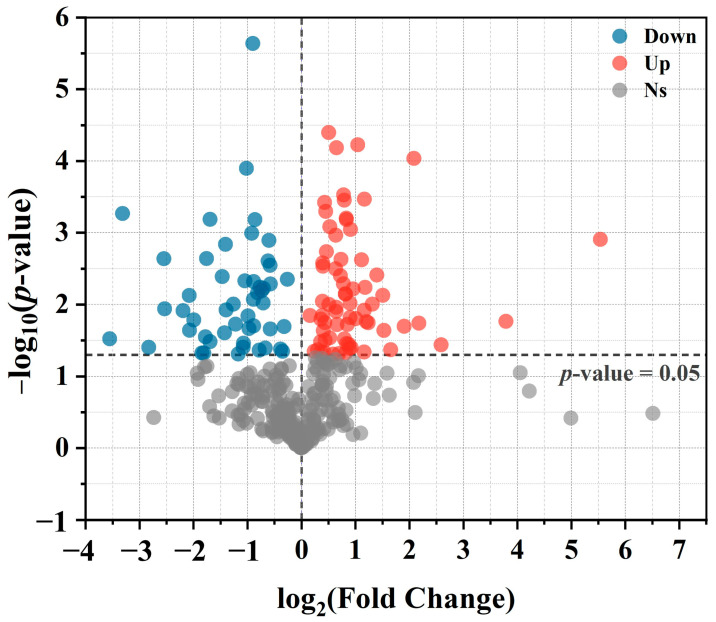
Volcano map of differential metabolites. Metabolites are displayed as points. The *X*-axis shows log_2_-transformed quantitative difference multiples of a metabolite between two treatments, and the *Y*-axis shows −log_10_ transformed *p*-value. Red dots represent upregulated differentially expressed metabolites, blue dots represent downregulated differentially expressed metabolites, and gray dots represent metabolites that are not significantly regulated (*p*-value < 0.05).

**Figure 5 microorganisms-12-00701-f005:**
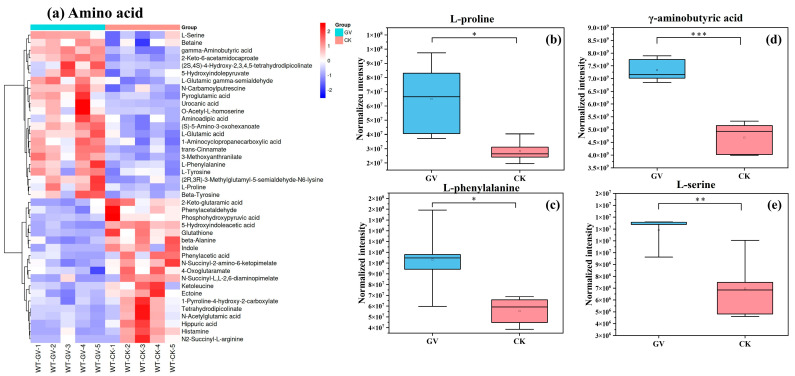
Hierarchical clustering heatmap of differential metabolites in amino acid category (**a**). The color scale indicates the relative levels of metabolites; red represents higher abundance, and blue represents lower abundance. Columns represent samples; rows represent metabolite names; “GV” refers to inoculated with AMF, *Glomus versiforme*; “CK” refers to no AMF inoculation treatment; “WT-GV” refers to *W. trilobata* inoculated with AMF, *Glomus versiforme*; “WT-CK” refers to *W. trilobata* not inoculated with AMF. On the left side of the heatmap, the clustering tree shows the hierarchical arrangement of differential metabolites. The box plot illustrates the relative expression levels of select key differentially expressed metabolites within the corresponding categories on the left, including L-proline (**b**), L-phenylalanine (**c**), γ-aminobutyric acid (**d**), L-serine (**e**). “*” denotes the significance of differences between the two groups: * *p* < 0.05, ** *p* < 0.01, and *** *p*< 0.001. The blank squares (“□”) in the boxplots are mean values.

**Figure 6 microorganisms-12-00701-f006:**
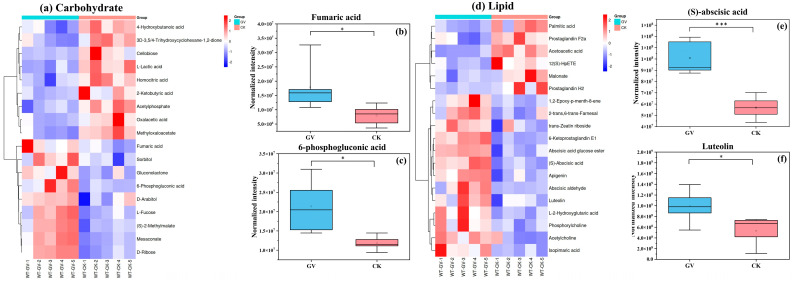
Hierarchical clustering heatmap of differential metabolites in carbohydrate (**a**) and lipid (**d**) category. The color scale indicates the relative levels of metabolites; red represents higher abundance, and blue represents lower abundance. Columns represent samples; rows represent metabolite names; “GV” refers to inoculated with AMF, *Glomus versiforme*; “CK” refers to no AMF inoculation treatment; “WT-GV” refers to *W. trilobata* inoculated with AMF, *Glomus versiforme*; “WT-CK” refers to *W. trilobata* not inoculated with AMF. On the left side of the heatmap, the clustering tree shows the hierarchical arrangement of differential metabolites. The box plot illustrates the relative expression levels of select key differentially expressed metabolites within the corresponding categories on the left, including fumaric acid (**b**), 6-phosphogluconic acid (**c**), (S)-abscisic acid (**e**), luteolin (**f**). “*” denotes the significance of differences between the two groups: * *p* < 0.05, and *** *p* < 0.001.

**Figure 7 microorganisms-12-00701-f007:**
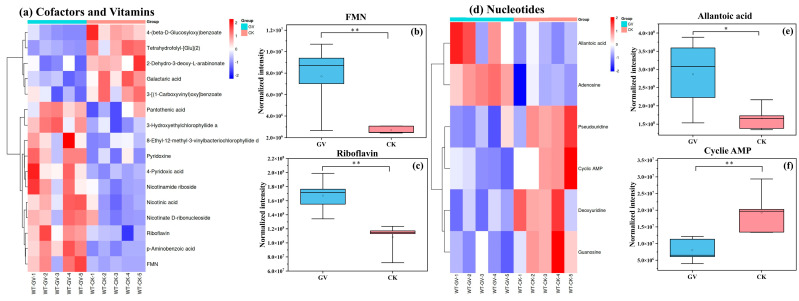
Hierarchical clustering heatmap of differential metabolites in cofactors and vitamins (**a**) and nucleotide (**d**) category. The color scale indicates the relative levels of metabolites; red represents higher abundance, and blue represents lower abundance. Columns represent samples; rows represent metabolite names; “GV” refers to inoculated with AMF, *Glomus versiforme*; “CK” refers to no AMF inoculation treatment; “WT-GV” refers to *W. trilobata* inoculated with AMF, *Glomus versiforme*; “WT-CK” refers to *W. trilobata* not inoculated with AMF. On the left side of the heatmap, the clustering tree shows the hierarchical arrangement of differential metabolites. The box plot illustrates the relative expression levels of select key differentially expressed metabolites within the corresponding categories on the left, including FMN (**b**), riboflavin (**c**), allantoic acid (**e**), and cyclic AMP (**f**). “*” denotes the significance of differences between the two groups: * *p*< 0.05, and ** *p* < 0.01.

**Figure 8 microorganisms-12-00701-f008:**
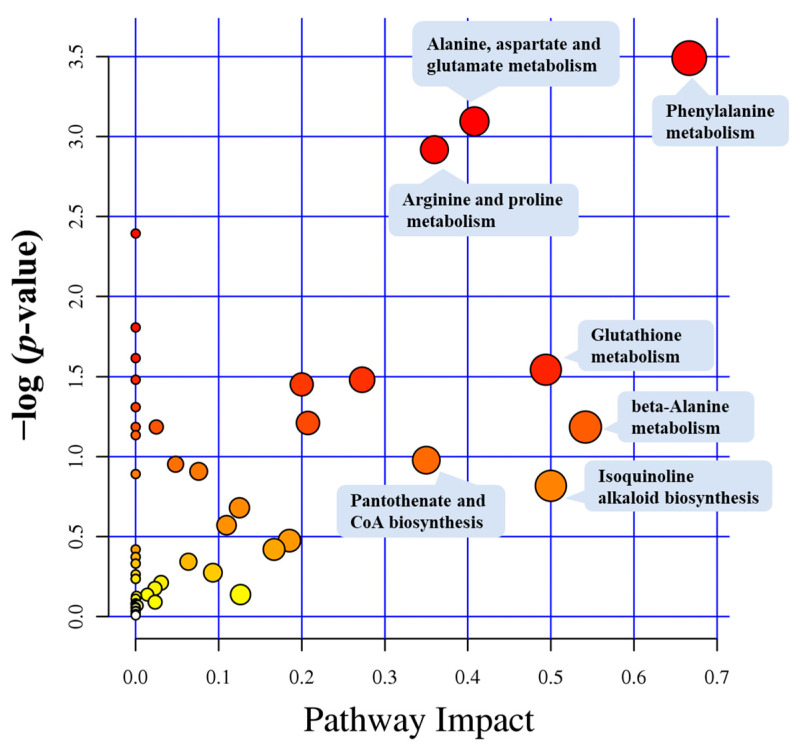
Metabolic pathway impact factor bubble plot for AMF inoculation treatments (*Glomus versiforme*, GV) compared with the treatments of no AMF inoculation (CK). Each bubble corresponds to a KEGG pathway. The *X*-axis shows the relative importance impact of metabolites within the pathway, and the *y*-axis indicates the pathway’s enrichment significance as −log_10_ (*p*-value). A higher Impact value corresponds to a larger bubble. Color intensity is linked to the *p*-value, with darker shades reflecting smaller *p*-values and lighter shades indicating larger *p*-values.

**Table 1 microorganisms-12-00701-t001:** The seven metabolic pathways most significantly influenced by AMF in *W. trilobata*.

Pathway ID	Description	Number	Metabolites	Kegg ID
ath00360	Phenylalanine metabolism	3	L-phenylalanine	C00079
			Trans-cinnamate	C00423
			Phenylacetaldehyde	C00601
ath00250	Alanine, aspartate and glutamate metabolism	5	L-glutamic acid	C00025
			2-keto-glutaramic acid	C00940
			Oxalacetic acid	C00036
			Fumaric acid	C00122
			γ-aminobutyric acid	C00334
ath00330	Arginine and proline metabolism	7	L-glutamic acid	C00025
			N-acetylglutamic acid	C00624
			L-proline	C00148
			L-glutamic gamma-semialdehyde	C01165
			Fumaric acid	C00122
			Oxalacetic acid	C00436
			γ-aminobutyric acid	C00334
ath00480	Glutathione metabolism	4	Glutathione	C00051
			γ-glutamylcysteine	C00669
			Pyroglutamic acid	C01879
			L-glutamic acid	C00025
ath00410	beta-Alanine metabolism	2	β-alanine	C00099
			Pantothenic acid	C00864
ath00950	Isoquinoline alkaloid biosynthesis	1	L-tyrosine	C00082
L-tyrosine	Pantothenate and CoA biosynthesis	2	β-alanine	C00099
			Pantothenic acid	C008640

## Data Availability

The data presented in this study are available on request from the corresponding author (e-mail: qishanshan1986120@163.com).

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
