# Peer review of "Impacts of Arbuscular Mycorrhizal Fungi on Metabolites of an Invasive Weed Wedelia trilobata"

_microorganisms, 2024, doi:10.3390/microorganisms12040701_

Round 1
Reviewer 1 Report (Previous Reviewer 2)
Comments and Suggestions for Authors
Comparative research on plant metabolomics is an interesting direction in the study of plant-microbial interactions, including those between plants and arbuscular fungi. The presented manuscript demonstrates high-quality experimental data.
Remarks:
Figures 5A, 6A, 6D, 7A, and 7D show the hierarchical clustering heatmaps of differential metabolites. However, I did not find a description in the Materials and Methods section of how these heatmaps were obtained. Because of this, I don't understand what the blue (-2) to red (+2) colors on these maps mean for each of the 10 plants in both treatments (control and AMF-treated). In the figure captions, the authors indicate, “The color scale indicates the relative amounts of metabolites. Red mean higher contents, blue mean lower contents.” What does “the relative amounts of metabolites” mean for each plant? For example, what do the red and blue colors for L-Serine in Figure 5A mean for WT-GV-1 and WT-CK-1 plants, respectively? Explain this in the Materials and Methods section. In addition, the figure captions do not explain what “GV” and “CK” mean.
Despite the positive impression of the manuscript, I do not understand why the authors submitted it to the journal Microorganisms. This work did not use microbiological methods, did not provide the characteristics of the microorganism – AMF (for example, what is the natural source of isolation of the AMF culture that the authors used?), and did not study different cultures of microorganisms. There are no microbiological topics in the manuscript. The authors use “arbuscular mycorrhizal fungi (AMF)” as a keyword. However, the manuscript shows changes only in plants without describing AMF. I find that Plant Biology (e.g., Plants) or Biochemistry (e.g., Metabolites) journals would be more appropriate for this manuscript.
Author Response
Response to Reviewer 1 Comments
Thank you very much for taking the time to review this manuscript. Please find the one-by-one responses below and the corresponding revisions and corrections highlighted in the re-submitted files.
Comments 1:
Figures 5A, 6A, 6D, 7A, and 7D show the hierarchical clustering heatmaps of differential metabolites. However, I did not find a description in the Materials and Methods section of how these heatmaps were obtained. Because of this, I don't understand what the blue (-2) to red (+2) colors on these maps mean for each of the 10 plants in both treatments (control and AMF-treated). In the figure captions, the authors indicate, “The color scale indicates the relative amounts of metabolites. Red mean higher contents, blue mean lower contents.” What does “the relative amounts of metabolites” mean for each plant? For example, what do the red and blue colors for L-Serine in Figure 5A mean for WT-GV-1 and WT-CK-1 plants, respectively? Explain this in the Materials and Methods section. In addition, the figure captions do not explain what “GV” and “CK” mean.
Response 1: In this experiment, agglomerative hierarchical clustering was presented, wherein each object is initially assigned to its own cluster and then merged into larger clusters successively until termination. The dataset was scaled using the “pheatmap package” in R (v3.3.2) software to generate a hierarchical clustering heatmap of metabolite relative quantification values.
In the heatmaps, columns represent samples, rows represent metabolite names. "GV" refers inoculated with AMF, Glomus versiforme; "CK" refers no AMF inoculation treatment; "WT-GV" refers to W. trilobata inoculated with AMF, Glomus versiforme; "WT-CK" refers to W. tri-lobata not inoculated with AMF. on the left side of the heatmap, the clustering tree shows the hierarchical arrangement of differential metabolites.
WT-GV-1 indicates the first sample out of five in the AMF inoculation treatment (GV), and WT-CK-1 represents the first sample out of five in the no AMF inoculation treatment (CK). The magnitude of metabolite relative abundance is depicted by different colors, with red indicating higher expression and blue indicating lower expression. For instance, in Figure 5A, the red color for L-serine in WT-GV-1 signifies higher expression compared to the non-AMF-inoculated sample, while the corresponding blue color for WT-CK-1 suggests lower L-serine expression relative to the AMF-inoculated sample. Our expression in the manuscript may not have been sufficiently clear, so we made revisions. (lines: 256-259, 270-275, 288-293, 308-313)
Comments 2:
Despite the positive impression of the manuscript, I do not understand why the authors submitted it to the journal Microorganisms. This work did not use microbiological methods, did
not provide the characteristics of the microorganism – AMF (for example, what is the natural source of isolation of the AMF culture that the authors used?), and did not study different cultures of microorganisms. There are no microbiological topics in the manuscript. The authors
use “arbuscular mycorrhizal fungi (AMF)” as a keyword. However, the manuscript shows changes only in plants without describing AMF. I find that Plant Biology (e.g., Plants) or Biochemistry (e.g., Metabolites) journals would be more appropriate for this manuscript.
Response 2: For why we chose Microorganisms, we provide the following reasons in the hope of addressing your concerns:
- Relevance of Topic: The study explores the impact of interactions between plants and AMF on the metabolites of the invasive weed Wedelia trilobata. This aligns with the scope of the journal Microorganisms which is focus on the interactions between microorganisms and plants.
- AMF is a significant subject of the study in microbiology, particularly due to its mutualistic symbiotic relationship with plants. Our study focuses on the influence of AMF on plant metabolites, directly relating to AMF's role in plant-microbe interactions.
- The study utilizes metabolomics to investigate the impact of AMF on plant metabolites, demonstrating the application of modern biotechnology in microbiology research. Metabolomics holds great meanings in the field of microbiology.
Therefore, we believe that our study aligns well with the scope of the journal Microorganisms.

Reviewer 2 Report (Previous Reviewer 1)
Comments and Suggestions for Authors
Dear editor, dear authors, thank you for the very improved version of the manuscript. I feel that the work was of high magnitude and the content is clear, as well as the scientific questions, main observations, and conclusions. However, I consider that some sections must be improved in the language. Following, I present my comments and suggestions.
L20, please change inoculation by “inoculated”.
L25, please use an acronym as DEM, which is also mentioned in L19 (use in this line the abbreviation).
L58-59. Please, take off “includes W. trilobata” (redundant).
L65-66. Please, to include directly in the text, for instance “…nutrients, mainly phosphorus (P)….”
L68. Please, consider changing “portion” by "organs".
L123-124, please, consider expressing as photoperiod, e.g. “...16/8 h day/night...”
L125-126. Usually, the P concentration in experiments using AMF is reduced. This is the case or not?
L131. Phillips and Hayman
L135-136. Please, consider changing this sentence... maybe "The fungal structures as hyphae, arbuscules and vesicles, were immediately observed under the microscope...”
L152. Please, use italics to differentiate centrifugal acceleration of grams.
L168-171. please, use past tense verb forms.
Legend Figure 1. Curiously, GV is not mentioned as the AM fungal species here used. Please, make some changes to include all the required information in the legend. The figures and tables must be self-explanatories. The same with the other legends along the results chapter.
L207. QC was previously defined in M&M. Use directly the abbreviation.
L210-217, consider the past tense.
Figure 2 legend. information about the experiment, the treatments and the abbreviation are missing. Please, consider that figures and tables must be self-explanatory.
L236, please, consider change this phrase... it is not clear. You are mixing singular and plural, but I think that the category is called unknown. Or is referred as others unknown categories?
L250-251. Please, modify this explanation. It is hard to read and understand.
L254-255. I guess this sentence is part of the previous sentence. If not, some elements are missing.
L267-268. If you take off this section it is clearer (maintaining the asterisks and equivalences). However, information about the abbreviations GV CK is missing.
L273. “…with growth magnitude second only to fumaric acid…”. please, modify the redaction.
Figure 6, please, consider changing the font size. Currently it is not possible to read even with zoom.
L283-284. Same comment as Fig. 5.
L295, in cofactors…
L299-300. Same comment as Fig. 5 and 6.
Figure 8. …but this enrichment is exclusively associated with mycorrhization as noted in the text. That explanation is not included here. It should be mentioned in the legend or in the figure itself, but it should be evident to which treatment these metabolic pathways respond.
L328. …other “plant” species…
L331-332. you must include some examples of the numerous studies...
L336. for me, this expression is "more" adequate than nutrient deprived. However, you supply with a nutrient solution presumably with all the nutrients in adequate concentration. I am not sure that in the current study the nutrient deficiency is present…
L336-337. Please, change by …plants AMF colonized…
L343. expansion is in italics.
L388. Speculate or suggest?
L406. plays an important role…
L434. ROS is not defined previously. In the text ahead it is not used either
L439. TCA is also not defined or used elsewhere in the manuscript. I suggest using the non-abbreviated text in both situations.
L471. Observed or evidenced?
Comments on the Quality of English Language
I am not a native english speaker, but clearly some sections must be improved. Please, see the comments regarding this situations.
Author Response
Response to Reviewer 2 Comments
Thank you very much for taking the time to review this manuscript. Please find the one-by-one responses below and the corresponding revisions and corrections highlighted in the re-submitted files.
Comments 1:
L20, please change inoculation by “inoculated”.
Response 1:Thank you for your valuable suggestion. I have already changed "inoculation" to "inoculated". (line: 20)
Comments 2:
L25, please use an acronym as DEM, which is also mentioned in L19 (use in this line the abbreviation).
Response 2:We have used” DEMs” in the text. (line: 19, 25, 189)
Comments 3:
L58-59. Please, take off “includes W. trilobata” (redundant).
Response 3:We have deleted “includes W. trilobata”. (Lines:56-59)
Comments 4:
L65-66. Please, to include directly in the text, for instance “…nutrients, mainly phosphorus (P)….”.
Response 4:We have revised the sentence according to your suggestion. (lines: 65-66)
Comments 5:
L68. Please, consider changing “portion” by "organs".
Response 5:We have revised this sentence. (line: 68)
Comments 6:
L123-124, please, consider expressing as photoperiod, e.g. “...16/8 h day/night...”.
Response 6:We have revised this sentence. (line: 124)
Comments 7:
L125-126. Usually, the P concentration in experiments using AMF is reduced. This is the case or not?
Response 7:Normally, the mycorrhization frequency is higher under low-P condition, and it is reported that AMF could enhance plant P uptake abilities. In this study, in order to ensure that AMF could play greater roles in the interaction with plant, low-P nutrition was applied to all the plants of these two treatments (control and AMF inoculation). That is, we used low-P of 1×Hoagland solution, containing 1.545 mg/L P. We have demonstrated this in the Materials and methods part. (Lines:125-126)
Comments 8:
L131. Phillips and Hayman
Response 8: We revised this. (line: 132)
Comments 9:
L135-136. Please, consider changing this sentence... maybe "The fungal structures as hyphae, arbuscules and vesicles, were immediately observed under the microscope...”
Response 9: We have made modifications to the original sentence. (lines: 136-137)
Comments 10:
L152. Please, use italics to differentiate centrifugal acceleration of grams.
Response 10:Thank you for your careful review. We have made the revisions. (line: 153)
Comments 11:
L168-171. please, use past tense verb forms.
Response 11:We have revised the manuscript. (lines: 169-176)
Comments 12:
Legend Figure 1. Curiously, GV is not mentioned as the AM fungal species here used. Please, make some changes to include all the required information in the legend. The figures and tables must be self-explanatories. The same with the other legends along the results chapter.
Response 12:In the "Experimental Design" and "Study Species" sections, we provided the detailed description of the AMF species we used: "AMF species, Glomus versiforme (GV)","AMF inoculation, with 6 g of GV fungal inoculum, uniformly mixed with 294 g of sand as the growth medium (AMF treatment, GV)" (lines: 107-108, 118-120, 204). Also, we added the detailed information in all the figure legends.
Comments 13:
L207. QC was previously defined in M&M. Use directly the abbreviation.
Response 13:We have abbreviated “quality control”. (line: 208)
Comments 14:
L210-217, consider the past tense.
Response 14:Based on your advice, we used the past tense. (lines: 210-218)
Comments 15:
Figure 2 legend. information about the experiment, the treatments and the abbreviation are missing. Please, consider that figures and tables must be self-explanatory.
Response 15:We have supplemented the information in the legend of Figure 2. (lines: 225-227)
Comments 16:
L236, please, consider change this phrase... it is not clear. You are mixing singular and plural, but I think that the category is called unknown. Or is referred as others unknown categories?
Response 16:Sorry for the confusing expression. There are 39 metabolites that have not been identified in the unknown category in Figure 3. Besides this, there is no other unknown categories any more. (lines: 238-239) We have revised this.
Comments 17:
L250-251. Please, modify this explanation. It is hard to read and understand.
Response 17:We have modified the legend description for Figure 4. (lines: 253-255)
Comments 18:
L254-255. I guess this sentence is part of the previous sentence. If not, some elements are missing.
Response 18:Thank you for your careful review. we have adjusted the expression order of the sentences. (lines: 256-259)
Comments 19:
L267-268. If you take off this section it is clearer (maintaining the asterisks and equivalences). However, information about the abbreviations GV CK is missing.
Response 19: We have supplemented more detailed information regarding GV and CK in the legends of Figures 5, 6, and 7. (lines: 269-275, 288-293, 308-313)
Comments 20:
L273. “…with growth magnitude second only to fumaric acid…”. please, modify the redaction.
Response 20:We have made modifications to the sentence. (lines: 279-282)
Comments 21:
Figure 6, please, consider changing the font size. Currently it is not possible to read even with zoom.
Response 21: All the graphs are using the required size pixels by this journal. The pixels would be better in the last publishing version.
Comments 22:
L283-284. Same comment as Fig. 5.
Response 22:We have supplemented more detailed information regarding GV and CK in the legends of Figures 5, 6, and 7. (lines: 269-275, 288-293, 308-313)
Comments 23:
L295, in cofactors…
Response 23:We have corrected this here. (line: 307)
Comments 24:
L299-300. Same comment as Fig. 5 and 6.
Response 24:We have supplemented more detailed information regarding GV and CK in the legends of Figures 5, 6, and 7. (lines: 269-275, 288-293, 308-313)
Comments 25:
Figure 8. …but this enrichment is exclusively associated with mycorrhization as noted in the text. That explanation is not included here. It should be mentioned in the legend or in the figure itself, but it should be evident to which treatment these metabolic pathways respond.
Response 25: We have supplemented the legend for Figure 8, which illustrates the metabolic pathway impact factor bubble plot obtained from the AMF inoculation group (Glomus versiforme, GV) compared to the no AMF inoculation group (CK), demonstrating the metabolic pathways influenced by AMF treatment. (lines: 335-336)
Comments 26:
L328. …other “plant” species…
Response 26:We have made modifications to the sentence. (lines: 345)
Comments 27:
L331-332. you must include some examples of the numerous studies...
Response 27:In this section, we have included references 33 and 34 to support our viewpoints. Later in the text, we provided some examples of how AMF promotes the growth of species like Solidago canadensis and Microstegium vimineum. (lines: 348-351)
Comments 28:
L336. for me, this expression is "more" adequate than nutrient deprived. However, you supply with a nutrient solution presumably with all the nutrients in adequate concentration. I am not sure that in the current study the nutrient deficiency is present…
Response 28:In this experiment, low-P nutrition was applied to all the plants of these two treatments (control and AMF inoculation). That is, we used low-P of 1×Hoagland solution, containing 1.545 mg/L P. We may not have elaborated on this aspect in detail in the manuscript, so we have provided additional information. (lines: 125-126)
Comments 29:
L336-337. Please, change by …plants AMF colonized…
Response 29: We have made modifications to the expression of the sentence. (line: 353-354)
Comments 30:
L343. expansion is in italics.
Response 30:We have corrected the italics for “expansion”. (line: 360)
Comments 31:
L388. Speculate or suggest?
Response 31:We have revised this. (lines: 405)
Comments 32:
L406. plays an important role…
Response 32:We have corrected this here. (line: 423)
Comments 33:
L434. ROS is not defined previously. In the text ahead it is not used either
Response 33:We have provided a comprehensive explanation of “ROS”. (line: 451)
Comments 34:
L439. TCA is also not defined or used elsewhere in the manuscript. I suggest using the non-abbreviated text in both situations.
Response 34:We have provided a comprehensive explanation of "TCA" and used non-abbreviated text in the manuscript. (line:456)
Comments 35:
L471. Observed or evidenced?
Response 35:We have replaced "observed" with "evidenced". (line: 489)
Comments 36:
I am not a native english speaker, but clearly some sections must be improved. Please, see the comments regarding this situations.
Response 36:Thank you for your efforts on this manuscript. We have made revisions to the manuscript according to your suggestions. Furthermore, the English text of this manuscript has been further refined by one of the authors, Misbah Naz, a native English speaker.

This manuscript is a resubmission of an earlier submission. The following is a list of the peer review reports and author responses from that submission.
Round 1
Reviewer 1 Report
Comments and Suggestions for Authors
Dear Editor, Dear Authors. I have had the opportunity to read the interesting article by Jiang et al., which has been of great interest to me; therefore, my appreciation is mainly positive about a possible publication, especially because it is a design that, despite being simple, provides valuable information about the role of AMF in aspects that ultimately lead to the metabolomic response of alien mycorrhizal plants that can influence their prevalence in environments in which they are present, and in competition with native plants. In addition, the tool used is novel in the study of this symbiosis and has been much less studied compared to other omics tools, especially transcriptomics. However, authors must perform several improvements before their manuscript is acceptable for publication, many of them in formal aspects.
The title should be more direct about what is expected as the goal of the study. "Metabolomics" is the omics tool that allows data to be obtained, but not the purpose of the research itself.
I didn't know the species, so I looked it up and most of the results referred to Sphagneticola trilobata. Please review the synonymy of the plants species and use the currently accepted term.
In the abstract it is named at first time as Wedlia. Please correct based on the correct name and the above comment.
Above-ground and below-ground are terms rarely used in this kind of study. Prefer Root and Shoot throughout the text.
Again in abstract, refer directly to the regulated ones (up or down). When it mentions that there are 119 "differential metabolites" it is understood that they are produced or not (presence or absence), when the correct thing to do is to refer to their differential concentration.
Please note in the abstract that comparisons were made in situations of AMF colonization with respect to non-colonized plants.
Dear authors, a relevant piece of information "always" when AMF inoculation is used is to report its level of colonization. In this case, this is relevant because it is possible to measure the degree of colonization as a determinant of the results obtained. Please include this relevant aspect in the abstract.
FORMAT. Separate citations from written text throughout the entire document.
Phosphorus can be abbreviated by its chemical symbol (P) saving space in the text. Use the same criteria for other elements or nutrients.
The genus Funneliformis has been widely accepted for at least a decade. Please use the currently accepted name and make changes accordingly in the text and figures, e.g. use (FV) instead of (GV).
Phyllips and Hayman are the authors of the classical method of staining, but the determination of colonization is not explained in that paper. What is here expressed is then a "frequency" of segments colonized by AMF, not the percentage of colonization. To refer to a percentage of colonization, a count must be made of points at which mycorrhization is observed, for example in a grid. Please refer to the term as mycorrhization frequency, and correct citations accordingly.
Dynamic exclusion "was" implemented to eliminate redundant MS/MS data...
What level of significance was used in the analysis of the data?
Microscopic (staining) analysis revealed an average colonization frequency of 81.03% across the examined root segments in AMF inoculated plants. (suggested as second phrase of first paragraph in page 5).
Figure 1. Information is missing from the legend. The figures must be self-explanatory, and in this case, there is no mention of the inoculated treatment and the control.
Page 6, last paragraph: (VIP ≥ 1 and P-value ≤ 0.05), a total… this is a point? If not, please rewrite.
Page 7, Legend of Figure 3. Log2 and log 10 are in basis, please use subscripts.
In figure 4 legend, there are no identification or explanation for the different subfigures. Please, re-do.
Page 9. My feeling in that the two first sentences in 3.3 are in fact methodology.
Page 11, first paragraph. Consequently, plants “colonized” with AMF typically manifest higher biomass in both aboveground and belowground tissues. Please change.
Page 11, first paragraph in section 4.2, please correct the grammar.
Page 11, third paragraph in section 4.2, the discussion about the proline is ambiguous. Please, indicate the environmental conditions argued.
Final paragraph in discussion: In summary, our experiments “have observed” the establishment of a close symbiotic relationship between W. trilobata and AMF. Have evidenced? Demonstrated? Revealed? The observation is an activity from the position of the researcher…
The conclusion is ambiguous, seems to be a summary of the study. Please, include actual conclusions in this section avoiding repeat a summary of results.
Reviewer 2 Report
Comments and Suggestions for Authors
The manuscript microorganisms-2774052 describes changes in the metabolomics of the plant Wedelia trilobata during the process of mycorrhization of roots by arbuscular fungi. In describing changes in metabolites, the authors provide data that may be of interest to a wide range of readers. However, the description of the methodology and discussion of the results require major revision before this manuscript can be recommended for publication.
Major remarks:
1. The description of methods and results should be more thorough. For example, in section 2.3 it should be indicated whether the level of colonization was determined for each plant and then the average for the sample (n = 5) was found, or whether the indicator was immediately determined for the entire variant. Section 3.1 shows the figure 81.03% (± how much?). Were the experimental plants without mycorrhizae or with negligible levels of colonization?
It is also necessary to describe in more detail in Section 2.4.1. “the sample” - What did the authors analyze?. Were these roots? Or leaves? Or stems? Or all parts of the plant at the same time? How was sample selection standardized? Because in different parts of the plant (or in different layers), differences in metabolism vary significantly. How old were the plants? How much time was between the points between which the authors compared the content of metabolites? This information is extremely important for understanding the results of the study and requires a detailed description.
2. In section 3.2.2, the authors indicate that they identified 119 metabolites with differential expression. However, Figures 4, 5 and 6 show statistical comparisons for only 16 metabolites. Were the differences between control and experiment statistically significant for the other 103 metabolites? This should be clarified due to the fact that even for the 16 metabolites described, the hot maps give ambiguous changes in metabolite concentrations among the five plants in the sample. The authors indicate that arbuscular mycorrhiza helps plants with water and mineral nutrition. Why don't the authors consider metabolites associated with these parts of metabolism? For example, ectoine.
3. In Section 4, I see some inconsistencies in the discussion of the results. On the one hand, the authors point to the growth-promoting activity of arbuscular mycorrhiza. On the other hand, the metabolites that change the most (what the authors call the “metabolic regulatory role of AMF”) are stress metabolites. And their increase indicates the activation of phytoimmunity, which is logical when a plant interacts with a fungus. However, defense mechanisms conflict with active growth. I did not see a discussion of metabolites that accelerate growth processes. How do increased concentrations of proline, GABA, and ABA promote faster plant growth?
Another contradiction I see is that the authors insistently remind that “fumaric acid is a key metabolite of the tricarboxylic acid cycle.” This is true, but it is also true that other metabolites of the TCA cycle do not change their concentrations to the same extent as fumaric acid. And fumaric acid is a central metabolite and is involved in many processes in the cell, including the metabolism of amino acids. Therefore, I do not understand why the authors appeal to the TsTK. Then analyze separately all the metabolites of the TCA cycle to understand whether it or its parts change their activity and direction of work.
Minor remarks:
1. The authors use the name for the plant Wedelia trilobata (L.) A.S. Hitchc., although the modern scientific name for this plant is Sphagneticola trilobata (L.) Pruski, 1996. The authors may have reasons for using an outdated species name, but this needs to be explained somehow in the “Introduction” section. And the abstract also begins with the incorrect spelling of the species name.
2. The phrase “metabolism pathway” should be avoided in the text without any explanation. What do the authors mean by the term “phenylalanine metabolism pathway”? Or "carbon metabolism"? What reactions correspond to these terms?
3. “indigenous to the tropical regions of South America.” - That's for sure? Other sources write, "to the tropical regions of Central America".
4. “now widely distributed across Asia, Africa” - describe more precisely in which parts of Asia and Africa. I have doubts that the described plant grows in the Sahara, Tibet, and Taimyr.
5. In the caption to Figure 1, instead of “different treatments”, decipher “GV” and “CK”. Also indicate what the bars mean. "Biomass" is wet weight? Was the dry weight of the plants determined? Perhaps the observed growth is related to the water supply of the plants?
6. Add line numbers to your manuscripts to make it easier to indicate minor errors, such as typos, in your review. So, “L-Proline, gamma-Aminobutyric acid, L-Phenylalanine, and L-Serine” should be written “L-proline, γ-aminobutyric acid, L-phenylalanine, and L-serine.”
Round 2
Reviewer 2 Report
Comments and Suggestions for Authors
1. Corrections in method descriptions are insufficient. Figures 4–7 show “quantitative difference multiples of a metabolite between two samples.” In a revised version of the manuscript, it was indicated that one of the samples were leaves of two-month-old plants (line 156). What is the second sample, in relation to which upregulation and downregulation were determined? I will repeat my question: How much time was between the points in which the authors compared the content of metabolites? It may be useful to provide the figure with a schema of the experiment indicating the duration of plant cultivation, inoculation with a mycorrhizal fungus, and the moments of sampling.
2. Lines 246-248: “These classes encompass amino acid (31.52%), lipids (21.74%), carbohydrate (14.67%), cofactors and vitamins (9.24%), nucleotide (6.25%), xenobiotics (4.89%), energy (0.82%), peptide (0.27%), and others unknow (10.60%) (Figure 3).” – This text completely duplicates the information in Figure 3. Indicate in the text of the manuscript, instead of percentages, the number of metabolites in each class.
3. Lines 207-209 and 483-484: “AMF exhibited a high level of colonization in the roots of W. trilobata. Microscopic staining analysis revealed an average mycorrhization frequency of 81.03% (± 6.45%) across the examined plants" and "our experiments observed a high mycorrhization frequency formed by AMF in W. trilobata", respectively. I did not find any discussion of these results in the Discussion section. On what basis do the authors claim that 81% of mycorrhization frequency is a high level of colonization? Provide a comparative analysis of the results obtained with the literature data
Author Response
Dear reviewer, thank you again for taking the time to review this manuscript. This is really helpful to improve our manuscript. Please find our point-to-point responses below and the corrections highlighted in the re-submitted files.
Comments 1:
Corrections in method descriptions are insufficient. Figures 4–7 show “quantitative difference multiples of a metabolite between two samples.” In a revised version of the manuscript, it was indicated that one of the samples were leaves of two-month-old plants (line 156). What is the second sample, in relation to which upregulation and downregulation were determined? I will repeat my question: How much time was between the points in which the authors compared the content of metabolites? It may be useful to provide the figure with a schema of the experiment indicating the duration of plant cultivation, inoculation with a mycorrhizal fungus, and the moments of sampling.
Response 1:In this experiment, we planted W. trilobata in only two treatments: 1) the non- AMF inoculation treatment (CK) and 2) the AMF-inoculated treatment (GV). There were 5 replicates for each treatment. (lines 129-133).
After two months growth, we harvested the second pair of fully expanded leaves from the top of each plant in the both two treatments at the same time for metabolomic analysis. We compared the differential metabolites between these two treatments, that is, were the metabolites induced by AMF inoculation different with the control treatment? The experiment design including only two treatments. We declared it clear in section 2.2 (lines 154-157).
Comments 2:
Lines 246-248: “These classes encompass amino acid (31.52%), lipids (21.74%), carbohydrate (14.67%), cofactors and vitamins (9.24%), nucleotide (6.25%), xenobiotics (4.89%), energy (0.82%), peptide (0.27%), and others unknow (10.60%) (Figure 3).” – This text completely duplicates the information in Figure 3. Indicate in the text of the manuscript, instead of percentages, the number of metabolites in each class.
Response 2:Thanks for your comments. We have indicated the quantity of each metabolite class in the manuscript. (lines 242-248)
Comments 3:
Lines 207-209 and 483-484: “AMF exhibited a high level of colonization in the roots of W. trilobata. Microscopic staining analysis revealed an average mycorrhization frequency of 81.03% (± 6.45%) across the examined plants" and "our experiments observed a high mycorrhization frequency formed by AMF in W. trilobata", respectively. I did not find any discussion of these results in the Discussion section. On what basis do the authors claim that 81% of mycorrhization frequency is a high level of colonization? Provide a comparative analysis of the results obtained with the literature data.
Response 3:In previous studies, the mycorrhization frequency of other species were lower than that of the results in our study. For example, the mycorrhization frequency for Solidago canadensis is about 20% (Qi, Wang et al. 2022); for Triadica sebifera is about 20% (Zhan, Xing, Ma et al. 2023); for some invasive plants are about 30% (Sun, Yang et al. 2022), Therefore, our experiment observed a higher level of colonization of AMF in roots of W. trilobata (about 80%). We have provided additional elucidation in the discussion section and incorporated references. (lines 388-340)
References:
Qi, S., J. Wang, L. Wan, Z. Dai, D. M. da Silva Matos, D. Du, S. Egan, S. P. Bonser, T. Thomas and A. T. Moles (2022). "Arbuscular Mycorrhizal Fungi Contribute to Phosphorous Uptake and Allocation Strategies of Solidago canadensis in a Phosphorous-Deficient Environment." Front Plant Sci 13: 831654.
Xing, Z., T. Ma, L. Wu, Z. Zhang, J. Ding and E. Siemann (2023). "Foliar herbivory modifies arbuscular mycorrhizal fungal colonization likely through altering root flavonoids." Funct Ecol 00: 1–13.
Sun, D., X. Yang, Y. Wang, Y. Fan, P. Ding, X. Song, X. Yuan and X. Yang (2022). "Stronger mutualistic interactions with arbuscular mycorrhizal fungi help Asteraceae invaders outcompete the phylogenetically related natives." New Phytol 236(4): 1487-1496.